# Optimization of Propagation of the Polish Strain of *Aldrovanda vesiculosa* in Tissue Culture

**DOI:** 10.3390/biology11101389

**Published:** 2022-09-23

**Authors:** Marzena Parzymies, Magdalena Pogorzelec, Alicja Świstowska

**Affiliations:** 1Institute of Horticultural Production, University of Life Sciences in Lublin, 20-950 Lublin, Poland; 2Department of Hydrobiology and Protection of Ecosystems, University of Life Sciences in Lublin, 20-950 Lublin, Poland

**Keywords:** photosynthetic pigments, cultivation media, disinfection of plant material, in vitro, media solidification, micropropagation, multiplication, regeneration, turion sterilization, waterwheel plant

## Abstract

**Simple Summary:**

*Aldrovanda vesiculosa* is a carnivorous aquatic plant. It floats in shallow water, where it preys on small organisms. Due to human activity and climate change, it is endangered all over the world, and without human intervention it may soon disappear. The article describes a method of in vitro propagation of *A. vesiculosa* making it possible to produce many plants, which can then be used to establish new populations of this rare and unique species or to strengthen existing ones.

**Abstract:**

*Aldrovanda vesiculosa* is a rare and critically endangered carnivorous plant species. Its populations have declined worldwide, so there is a need to protect the species from extinction. The research was conducted to establish an effective method of in vitro propagation of the species in order to obtain plants for reintroduction in the wild. The procedures included disinfection, multiplication, and acclimatization of plants. Contamination-free in vitro cultures were established using shoots and turions, which were disinfected with 0.25% sodium hypochlorite. The shoots were first defoliated. The explants regenerated better in liquid 1/5 MS medium than in solidified one. The optimum medium for the multiplication phase contained MS macro- and microelements diluted to 1/10. Plants cultivated in that medium were of good quality, long, and branched. The advantageous effect of medium was also confirmed by the content of photosynthetic pigments in the plant material. The content of chlorophyll *a* was highest in plants cultivated in 1/5 or 1/10 MS medium. The plants obtained were acclimatized to ex vitro conditions and reintroduced in the wild.

## 1. Introduction

*Aldrovanda vesiculosa* L., known as the waterwheel plant, is a very rare and critically endangered aquatic carnivorous perennial plant [1]. *Aldrovanda* is a monotypic genus containing only one species: *A. vesiculosa* L. It belongs to the Droseraceae family and the order Nepenthales [2,3]. As a Tertiary (Paleogene) element, *A. vesiculosa* is recognized as a relict species [1,4].

*A. vesiculosa* is rootless and floats below the water surface [2,5,6]. It produces a poorly branched shoot up to 20 cm long. A thin stem bears leaf whorls with 6 to 9 leaves each, terminating in traps. Each trap consists of a two-lobed lamina with a midrib and 3–6 long bristles [2,7]. According to Poppinga et al. [8], the traps close rapidly, within only 10 ms, when mechanically irritated by prey in warm water. The rapid closure of traps is caused mainly by turgor changes in the trap near the midrib, but a release of energy stored in the bent midrib is also possible [9].

Temperate *Aldrovanda* populations can flower and set seeds under favorable conditions, but this is rarely observed. The seeds have a limited thermal window and are capable of germinating predominantly at 25 °C [10]. The plants propagate mainly vegetatively, by shoot branching [7,11,12]. Adult plants may form up to eight branches, but the usual number is two to four; many individuals do not produce any branches at all. All branches separate from the mother shoot to form a new plant [10,12]. According to Kamiński [7], as well as Adamec and Kovarova [13], branching ability indicates optimal ecological habitat conditions.

As a response to unfavorable conditions, waterwheel plants form turions, which are a kind of vegetative dormant winter buds. The turions are dense leaves growing at highly shortened internodes that surround and protect fragile summer shoots from freezing [14]. All aldrovanda specimens growing in temperate climate conditions form turions, but these organs occur even in some Australian populations [2,12,14,15,16,17].

*Aldrovanda* grows in various habitats in shallow, standing, or very slow-flowing dystrophic waters [2]. The optimal stands are small bays or depressions in lake littoral zones, within or near tall emergent vegetation, which forms extensive, favorable niches. It protects plants against unfavorable winds or water waves. Falling leaf litter is another important factor for maintaining optimum water chemistry [2,18].

*Aldrovanda* has recently attracted the attention of nature conservationists due to its decline in many sites [19,20]. It is recognized as endangered (EN) under criteria B2ab (iii,iv,v), with a decreasing population trend [20]. The species is also recorded in Appendix I of the Bern Convention [21]. In Poland, it is listed on the Polish Red List of pteridophytes and flowering plants [22].

In the past, *A. vesiculosa* covered a vast territory, including all continents of the Old World, across various climatic zones [5]. At present, it is considered extremely scarce and is extinct in many countries and regions. There are about 50 extant, verified sites of the species worldwide on four continents [2,5,20]. Recent data indicate that Europe, mainly central and eastern, is the largest reservoir of the world’s *A. vesiculosa* individuals. At least 79 historical sites of the species were recorded in Poland in the last 200 years [7,11,23,24], but only nine natural sites were verified and confirmed between 2006 and 2013 [23,24].

The biology and ecology of *A. vesiculosa* in Europe are quite well known. The plants have been studied in natural sites and in outdoor cultures in Poland and the Czech Republic [7,23,25,26]. The Polish *A. vesiculosa* population was also used to introduce the species to potentially suitable sites in Central Europe, in order to protect the vanishing European population [8,9,15,16,18,23,27,28,29,30].

Mass propagation and reintroduction into natural habitats may be an effective way to conserve the declining populations of species which are at risk of extinction [31,32], mainly due to anthropogenic pressure causing habitat loss, fragmentation, overexploitation, pollution and climate change [33,34]. While conservation activity should focus primarily on protecting populations in their natural habitats (in situ) [35], ex situ techniques may be used to complement them. In the case of some species, it may be the only option [36]. The Global Strategy for Plant Conservation (2002) provides a framework for achieving harmony between ex situ conservation methods and associated in situ conservation of rare and vulnerable species [37]. There is an increasing need to integrate in situ and ex situ conservation methods in order to optimize the effects [20]. The importance of ex situ conservation has received international recognition with its inclusion in Article 9 of the Convention on Biological Diversity (CBD) and Target 8 of the Global Strategy for Plant Conservation (www.bgci.org.uk/files/7/0/global_strategy.pdf, accessed on 30 August 2022).

Currently, plant tissue culture plays an important role in the conservation of rare and endangered species [38]. In vitro propagation is a highly effective method for rapid, large-scale cloning of individuals from limited plant material in controlled environmental conditions [39]. Tissue culture techniques were used for propagation and conservation of many endangered plant species, especially those difficult to propagate by classical methods [40,41,42]. Kułak et al. [43] noted that in vitro methods are useful in protection and preservation, sustainable management, restoration, and rewilding. An integral element of in vitro conservation is producing plants in large enough numbers for reintroduction in the wild [44].

Ex situ conservation using biotechnological methods requires the establishment of methods for collection and disinfection of the plant material, regeneration and conservation of the plant species, and evaluation of its genetic stability [37].

Plant tissue culture is usually initiated in the laboratory after being collected in the field. In the case of many endangered or rare plant species, the amount of available plant material is very small. Seeds are preferred for in vitro culture, as they provide the largest genetic base, but for some species they are not available, and then vegetative organs are used [40].

The plant material must be disinfected, as contamination of exogenous and endogenous origin is the main problem for initiation of in vitro culture of plants. Plant material is usually initiated into an axenic culture by various sterilization procedures [36]. Contamination from internal sources are especially problematic, as many plants contain endophytic fungi or bacteria [45].

The next step, after disinfection, is the induction and multiplication of shoots. Many plant species have specific in vitro requirements for multiplication so there is a need to establish a suitable media formulation to for regeneration and growth of regenerants. Similarly, the physical conditions of culture maintenance should be customized for the species [36]. Appropriate methods of in vitro culture should be chosen, avoiding methods which might induce somaclonal variations, such as callus or suspension cultures [40,46].

Once the desired number of regenerants is obtained, successful rooting and acclimatization of plants are fundamental to the in vitro culture of the threatened species. Rooted and hardened plants may be used for further purposes, such as reintroduction into the wild or establishing collections in botanical gardens [36].

In vitro techniques were proven useful in the propagation of a large number of threatened plants. They were successfully used to protect such species as *Leopodium alpinum* [47], *Prosopis cineraria* [48], *Androcalva perlaria* [49], *Primula veris* [50], *Primula farinosa* [51], or *Salix lapponum* [52,53,54].

Although in vitro methods for the conservation of endangered plant species are widely used, information on in vitro propagation of *A. vesiculosa* is fairly scarce. It was propagated in vitro through shoot proliferation, but the multiplication rate was limited and not sufficient for large-scale plant production. These studies describe propagation of the Japanese strain, as disinfection of the Polish strain proved difficult [55,56].

The aim of this study was to present an efficient micropropagation protocol for in vitro propagation of the Polish strain of *A. vesiculosa*.

## 2. Materials and Methods

The plant material comprised *Aldrovanda vesiculosa* individuals collected from four peat bog lakes situated in East Poland: Lake Długie (51°27′17.18″ N; 23°10′12.60″ E) and Lake Łukie (51°24′40.30″ E; 23°04′56.73″ E) in Poleski National Park, and Lake Orchowe (51°29′27.63″ N; 23°34′26.12″ E) and Lake Płotycze (51°23′38.42″ N; 23°36′59.84″ E) in Sobiborski Landscape Park. The shoots were collected several times from each stand during the growing period, in July. The turions were collected several times in January and February, when they were buried in silt. The shoots and turions used as a source of explants are presented in Figure 1 and Figure 2. The shoots and turions were transported to the laboratory in plastic containers containing water from the lakes.

### 2.1. Disinfection of Plant Material

The shoots were disinfected immediately upon arrival at the laboratory. First, they were cut into smaller pieces about 3 cm long and defoliated. Then, they were rinsed under running tap water and shaken in water with a drop of detergent (Ludwik, GRUPA INCO S.A., Warsaw, Poland) twice for 10 min, followed by immersion in 70% ethanol (Chempur, Piekary Śląskie, Poland) for 10 s. Next, surface sterilization was performed by shaking the shoots in sodium hypochlorite (NaOCl, Chempur) at concentrations of 0.25%, 0.5%, or 1% for 5 min. The disinfected shoots were rinsed three times in sterilized distilled water and placed in 100 mL Erlenmeyer flasks containing 1/5 MS medium, which consisted of 5-fold diluted Murashige and Skoog (MS) [57] macro- and microelements. It was supplemented with 0.1 mg·dm^−3^ thiamine (vit. B_1_), 0.5 mg·dm^−3^ pyridoxine (vit. B_6_), 0.5 mg·dm^−3^ niacin (vit. PP), 2.0 mg·dm^−3^ glycine, 100 mg·dm^−3^ myo-inositol, and 20 g·dm^−3^ sucrose. Medium was selected based on preliminary observations. Medium pH was established at 5.5. Half of medium was solidified with 6.75 g·dm^−3^ of agar, and half was left liquid. Medium was steam-sterilized for 21 min at 121 °C under 1 hPa pressure. There were 50 shoot fragments used per treatment.

After being brought to the laboratory, turions were placed in a refrigerator at 4 °C. In March, after 49–70 days, half of them were placed in water and kept in a growing room at 18 °C with a 16-h photoperiod. The other half were disinfected, i.e., rinsed under tap water, shaken in water with a drop of Ludwik detergent, dipped in 70% ethanol, and then surface-sterilized with NaOCl 0.25% for 5 min. The disinfected turions were placed in liquid 1/5 MS medium. There were 20 turions used per treatment.

The flasks were numbered, with different numbers assigned to possible clones (explants obtained from one mother plant or turion). These numbers were retained during the entire cultivation process. The flasks with explants were placed in a growing room at 25 °C, with a 16-h photoperiod. The source of light was fluorescent, day-light lamps, with 30 µmol·m^−2^·s^−1^ light intensity (Philips Master TL-D 58W/840). The experiment lasted for 3 months. Observations concerning the contamination and regeneration rates were made once a week. Finally, the number of explants without symptoms of contamination, the number of regenerating explants, and the number of explants with signs of necrosis were counted. The regenerated shoots were placed in fresh media of the same composition. Passages were completed every 6 weeks. Each time, only the green parts were used.

### 2.2. Influence of the Media Type on the Growth of A. vesiculosa Shoots

The plant material for the experiment comprised shoots of *A. vesiculosa* obtained from a stabilized culture. The shoots were placed in 450 mL jars containing 150 mL of the following media: 1/5 MS (Murashige and Skoog diluted 5 times), 1/5 B_5_ (B5 Gamborg [58] diluted 5 times), 1/2 MS with KNO_3_ lowered to 500 mg·dm^−3^, and 1/2 B_5_ with KNO_3_ lowered to 500 mg·dm^−3^. All the media were liquid and contained 2% sucrose, with pH set up to 5.5 before autoclaving. The jars were covered with transparent screw caps. The jars with plants were placed in the growing room at 25 °C with a 16-h photoperiod. The light intensity was 30 µmol·m^−2^·s^−1^. The experiment lasted for 6 weeks, after which the shoots were measured and placed in fresh media of the same content for another 6 weeks. The following features were estimated after each passage: number of regenerating explants, length of the main shoot, number of explants with lateral shoots, number of lateral shoots, and length of lateral shoots.

### 2.3. Influence of the Media and Nitrogen Concentration on the Growth of A. vesiculosa Shoots

The plant material for the experiment comprised shoots of *A. vesiculosa* derived from a stabilized culture. The explants were shoot tips 2 cm long. The explants were placed in 450 mL jars containing 200 mL of MS, 1/2 MS, 1/5 MS, or 1/10 MS medium with 2N, 1N, or 1/2N (where N was KNO_3_ + NH_4_NO_3_; at a concentration doubled, unchanged, or halved in relation to the MS formulation). The media were left liquid. The pH was set up to 5.5 before autoclaving. There were 4 jars per treatment, and each jar contained 10 explants. The experiment lasted for 8 weeks, after which the following features were estimated: percentage of regenerating explants, length of the main shoot, fresh weight of the main shoot, percentage of branching explants, number of side shoots, and their length and fresh weight.

### 2.4. Content of Photosynthetic Pigments in A. vesiculosa Leaves

The content of assimilation pigments—chlorophyll *a*, *b* and carotenoids—was determined. The plants were weighed at the end of the experiment and ground in an agate mortar in 1 mL of 96% ethyl alcohol. The ground material was transferred to Eppendorf tubes, which were then placed in a water bath for 5 min at a temperature below the boiling point of the alcohol (70–73 °C). Then, the test tubes were placed in an MPW–350 R centrifuge. The extract was centrifuged for 10 min at 14,000 rpm (18,228 RCF) at 17 °C. The supernatant was cooled, and the absorbance was measured on a Specord 40 spectrophotometer at three wavelengths: 665, 649, and 470 nm. The contents of chl*a*, chl*b*, and total carotenoids in the fresh weight of plants were calculated according to modified formulas proposed by Lichtenthaler and Wellburn [59]. There were three repetitions for each treatment.

### 2.5. Acclimatization and Reintroduction Process

The plants were rinsed under tap water and placed in plastic containers of 50 × 30 × 12 cm (length × width × depth), filled with distilled water, and filtered lake water in a 1:1 ratio (*v*/*v*). Each clone was placed in a separate container (Figure 3). The containers were placed in a growing room, under the same conditions as during in vitro cultivation.

After 2–3 weeks, the shoots were transported in plastic containers to outdoor tanks at the University research station at Lake Piaseczno, situated at the boundary of Poleski National Park. The tanks were 79 × 47 × 30 cm in size (length × width × depth) with a volume of 90 l. The tanks were filled with lake water up to 2/3 of the depth. Clumps of *Carex* ssp. were also placed in the tanks. The tanks were covered with a net to keep seeds of other plant species away from the water, to avoid transporting them later to the wild. The tanks were equipped with an automatic water filling system with floats (Figure 4).

After about 4 weeks, depending on weather conditions, the plants were transported to natural lakes situated in Poleski National Park and its surroundings and placed in the water—5000 individuals per lake. In each lake, about 500 of the plants were placed in specially prepared 100 × 50 × 40 cm nets (length × width × depth) to facilitate monitoring of their condition (Figure 5).

### 2.6. Statistical Analysis

Where relevant, statistical analysis of the data was performed with Arstat software (University of Life Science in Lublin, Poland), using one-way or two-way ANOVA for one- or two-factorial design. The significance between means was estimated with Tukey’s confidence intervals at a 5% level of significance.

## 3. Results and Discussion

*Aldrovanda vesiculosa* is a plant species threatened with extinction. Poland is the largest reservoir of *Aldrovanda* plants, but the number of populations of the species and the number of individuals decreased. For this reason, we decided to take measures leading to restitution of the species in eastern Poland. Our research aimed to obtain *A. vesiculosa* plants and use them in the reintroduction process. We decided to use in vitro propagation because the availability of plant material is very low, and micropropagation makes it possible to obtain a large number of progeny from a limited number of mother plants.

### 3.1. Disinfection of Plant Material

The first challenge was to initiate a tissue culture. Our initial attempts to disinfect the shoots failed. Kondo et al. [56] pointed out that shoot tips are difficult to sterilize because they contain numerous aquatic microorganisms, such as bacteria, viruses, and water molds, so they decided to use winter buds (turions) instead. They used 1% benzalkonium chloride for 5 min, 1–2% sodium hypochlorite for 5 min, and 70% ethanol for a few seconds.

In our study, half of the turions were placed in water, at 22 °C, and these started grow. The other half were disinfected and inserted in the 1/5 MS media. The disinfection rate was very high, as 90% of explants were contamination-free. However, only 20% regenerated into shoots. Turions are available only during the winter and are difficult to obtain, so we decided to use shoots as well. Based on the experience of Kondo et al. [56], we removed the traps, which contained lake water. We used sodium hypochlorite at three concentrations: 0.25%, 0.5%, and 1%. The concentrations of 0.5% and 1% was too strong, as all the explants became transparent and died.

The disinfected shoot explants were placed in liquid or solidified 1/5 Murashige and Skoog medium (1962). The regeneration rate was found to be higher in liquid medium (68%) than in solidified medium (3.5%), as shown in Table 1.

On the basis of visual observation, it was concluded that shoots cultivated in liquid medium formed traps typical for the species, while those on solidified medium had shortened internodes and no traps (Figure 6).

Liquid medium, which seems to be a natural choice for an aquatic plant, was used by Kondo et al. [56], Nitta [60], and Parzymies et al. [61] for in vitro propagation of *A. vesiculosa*. Other studies demonstrated that water plants can be successfully cultivated in solidified cultivation medium [62,63,64]. Solidified medium was also used for the species of the Droseraceae family, such as *Dionea muscipula*, which is genetically related to *A. vesiculosa* [65].

### 3.2. Influence of the Media Type on Growth of A. vesiculosa Shoots

Following the initiation of the *A. vesiculosa* tissue culture, the contamination-free regenerated shoots were placed on fresh media at 6-week intervals. The stabilized shoots were used to set up an experiment to estimate the influence of the media type on the growth of *A. vesiculosa* shoots in vitro. The media formulations were 5-fold diluted MS (1/5 MS) or Gamborg B5 (1/5 B5) and twice-diluted MS and B5 media with KNO_3_ content lowered to 500 mg·dm^−3^ (1/2 MS KNO_3_ 500 and 1/2 B5 KNO_3_ 500, respectively).

All explants cultivated in 1/5 MS medium began to regenerate, and after 6 weeks they were 76.80 mm long (Table 2). On that medium, all shoots formed lateral shoots as well. On the remaining media, we obtained 60% growing shoots. The addition of KNO_3_ to the MS or B5 media decreased the length of the main shoot. Representative shoots are shown in Figure 7.

The results did not confirm the findings of Adamec and Pásek [55], who noted that B5 medium with the addition of only 500 mg·dm^−3^ of KNO_3_ was optimal for *A. vesiculosa* micropropagation. Those authors, however, used the Japanese strain of the species for in vitro cultivation. The Polish strain was established in tissue culture by Kondo et al. [56] using of B5 liquid medium, but the plants did not differentiate into multiple shoots, shoot primordia, or other cultures.

The shoots were moved to fresh media of the same content. Again 1/5 MS medium resulted in longer shoots, in comparison to other media (Table 3). The longer the shoots, the more secondary explants were obtained.

As the 1/5 MS media appeared to be more advantageous, we decided to conduct further experiments to establish which concentration of Murashige and Skoog medium would be optimal for the growth and development of *A. vesiculosa* shoots. The change in KNO_3_ content was disadvantageous for the regeneration and branching of shoots, so we decided to estimate the influence of the decrease in nitrogen concentration as well.

### 3.3. Influence of the MS Media and Nitrogen Concentration on the Growth of Shoots

All shoots cultivated in the 1/2MS 1/2N, 1/5MS 1/2N, 1/10MS 2N, 1N, and 1/2N media were observed to undergo further growth. The shoots in the MS 2N and 1N media turned brown and died (Table 4).

The longest shoots were obtained in 1/10MS 1N medium (35.64 mm) (Table 4). Shoots of similar length were observed in 1/10MS and 2N (30.85 mm) and 1/5MS salt and 1N (31.77 mm). The length of the shoots increased as the concentration of MS macro- and microelements decreased. A similar tendency was observed for the concentration of nitrogen compounds.

The highest percentage of plants with lateral shoots was obtained in 1/5MS 1/2N medium (79.5%), followed by the 1/10MS 1N (66.7%) and 1/5MS 1N (60%) media (Table 5).

The lowest number of branched shoots was noted in the 1/5MS 2N medium (25.6%). The number of lateral shoots was similar for all treatments.

Shoots obtained in different media are shown in Figure 8.

The length of the shoots and the number of lateral branches determine the multiplication rate of shoots in tissue culture, so they may be considered the most important parameters for regenerating explants. According to Kamiński [7], as well as Adamec and Kovarova [13], branching ability indicates optimal ecological habitat conditions, which suggests that it may also be applied to the condition of plants cultivated in vitro. Taking into consideration the number of regenerating plants, the number of branching plants, and the length of the main and lateral shoots, we recommend 1/10MS 1N medium formulation as optimal for in vitro cultivation of *Aldrovanda vesiculosa* plants. That medium composition was also used for micropropagation of the same species by Nitta et al. [60], who observed that the shoots proliferated and formed axillary buds.

The preference of *Aldrovanda vesiculosa* for a culture medium low in macro- and microelements may be explained by the fact that carnivorous plants often occur in natural sites low in nutrients, especially N and P [12,16,66]. Carnivorous plant species are able to supplement the low nutrient availability in the environment by absorbing nutrients from the prey captured in their traps, which can meet up to 100% of seasonal N and P needs [16]. Aquatic carnivorous plants take up nutrients directly from water with their floating shoots or from caught prey. However, the efficiency of foliar mineral uptake of aquatic carnivorous plants is still unknown [66].

Carnivorous plants are a phenomenon in the plant kingdom. Due to their low propagation rate in their natural environment, they are often propagated in vitro [67]. There is limited data regarding micropropagation of *A. vesiculosa*. However, in vitro propagation of species of the Droseraceae family was studied by many researchers around the world, who tested various solidified and liquid media and growth regulators. 1/2MS and RM media were optimal for in vitro propagation of plants of the genus *Drosera* [67]. Jang et al. [68] tested shoot proliferation of *Drosera* spp. using different concentrations of macroelements in MS medium. He obtained the best results for 1/3 MS medium, and the worst for 1/9 MS. According to Grevenstuk et al. [69], the best concentration for *Drosera intermedia* was 1/4 MS. The same medium was optimal in the case of *D. rotundifolia* [70]. Rejthar et al. [71] observed optimal in vitro growth of *D. intermedia* when the concentration of MS medium was reduced to 1/8.

### 3.4. Content of Photosynthetic Pigments in A. vesiculosa Leaves Depending on the Media and Nitrogen Concentration

There is little information available concerning the use of photosynthetic pigments to determine the condition of carnivorous plants, as measured by their photosynthetic efficiency. Adamec [28] reported that feeding on prey had no significant influence on chlorophyll content in the leaves of *A. vesiculosa*. Pavlovic et al. [72], investigating the effect of increased nitrogen intake from prey on the photosynthetic rate in *Nepenthes talangensis*, found that greater nutrient absorption increases the photosynthetic efficiency of leaf blades. He also argued that an increased rate of photosynthesis improves the growth and reproduction of plants and is likely to strengthen the competitive advantage of carnivorous over non-carnivorous species in nutrient-poor habitats.

Little is known about traditional uptake of nitrogen compounds from the environment by insectivorous plants, in the form of absorbable ions, and the importance of this process for metabolic processes. We compared the content of the photosynthetic pigments in plants grown in vitro on MS medium (dilutions: MS, 1/2 MS, 1/5 MS, 1/10 MS), with different amounts of available nitrogen, and plants from the natural habitat, where the N content was close to 0 (approximately 0.004 mg·dm^−3^). The highest content of chlorophyll *a* (chl*a*) and the highest sum of photosynthetic pigments in fresh weight were noted in plants cultivated in the media 1/5MS 1N (950.41 µg·g^−1^ and 1680 µg·g^−1^, respectively) and 1/10MS 1N (1067.2 µg·g^−1^ and 1817 µg·g^−1^). The values exceeded those established for plants derived from natural habitats (711.82 µg·g^−1^ and 1460 µg·g^−1^, respectively). Plants grown on MS or 1/2 MS media had a much lower total content of photosynthetic pigments (<1000 µg·g^−1^), but also high carotenoid content compared to other pigments (especially in the case of MS 1/2N and 1/2MS 2N) (Figure 9, Appendix A).

Carotenoid content was also relatively high in the plants from the natural habitat. With the low chl*a*:chl*b* ratio (1.8), this could indicate that plants were older than those grown in the laboratory. The chlorophyll *a*:*b* ratio in the fresh weight of plants cultivated in vitro ranged between 2.1 and 2.7. Only in the plants cultivated in the 1/2MS media with a double concentration of nitrogen (2N) it was lower (1.8).

Analysis of the content of photosynthetic pigments in the fresh weight of plants cultivated in vitro revealed a general trend indicating that the plants best predisposed to efficient photosynthesis (with the highest content of chlorophyll *a* in fresh weight) were those growing on media with lower content of mineral salts (1/5MS and 1/10MS), but also with a balanced nitrogen content in medium (1N) (Figure 9, Appendix A).

### 3.5. Acclimatization and Reintroduction Process

The *A. vesiculosa* cultures were passaged at 6-week intervals in the 1/10 MS media. After 5 cycles, a multiplication rate from 2.8 to 9.9 (5.22 on average) was obtained (Figure 10).

The plants were placed successively in plastic containers, and all of them underwent acclimatization to ex vitro conditions. When the back of the plant began to turn brown, the plants were transported to outdoor tanks to adapt to natural climate conditions.

The plants were kept in the outdoor tanks for about 4 weeks, after which they were transported to natural lakes. To facilitate monitoring of the plants’ condition, in each lake 500 of the 5000 individuals were placed in floating nets. The reintroduced plants will be monitored for the next few years.

## 4. Conclusions

Micropropagation of *A. vesiculosa* may make it possible to save the species from extinction. We present a micropropagation protocol method for the Polish strain of *A. vesiculosa*.

To establish a tissue culture, shoots, or turions may be used. Before disinfection, shoots should be defoliated to eliminate contaminants present in the traps. The explants can be surface-sterilized by rinsing them under tap water, shaking in water with a drop of detergent twice for 20 min, immersing in 70% ethanol, and shaking in 0.25% sodium hypochlorite (NaOCl) for 5 min. The explants regenerated better in liquid 1/5 MS medium than in solidified one. The optimal medium composition for large-scale in vitro propagation of *A. vesiculosa* shoots, elongation of the main shoot and the formation of lateral shoots was liquid Murashige and Skoog medium diluted 10 times (1/10 MS). The shoots passaged at 6-week intervals had a multiplication rate from 2.8 to 9.9. The favorable effect of the media was confirmed by the content of photosynthetic pigments in the plant material. The content of chlorophyll *a* in the fresh weight was highest in the leaves of plants cultivated in 1/5 or 1/10MS medium. The in vitro plants were acclimatized to ex vitro conditions and reintroduced into the wild.

## Figures and Tables

**Figure 1 biology-11-01389-f001:**
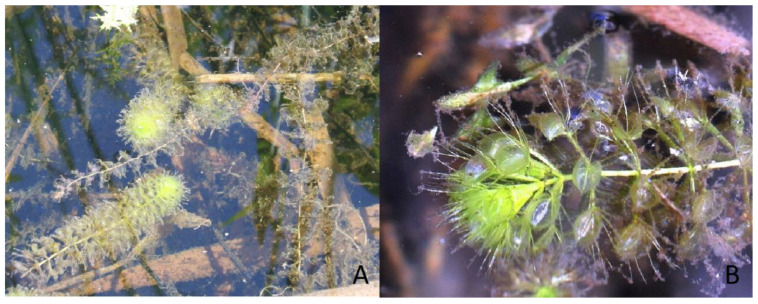
(**A**,**B**) *Aldrovanda vesiculosa* plants floating below the water surface.

**Figure 2 biology-11-01389-f002:**
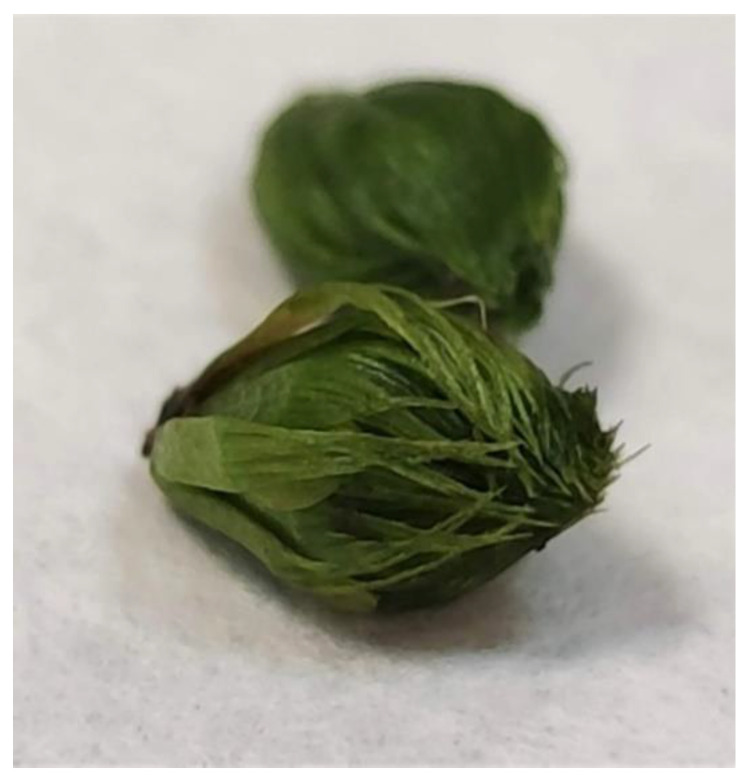
Turions of *Aldrovanda vesiculosa* used for tissue culture initiation.

**Figure 3 biology-11-01389-f003:**
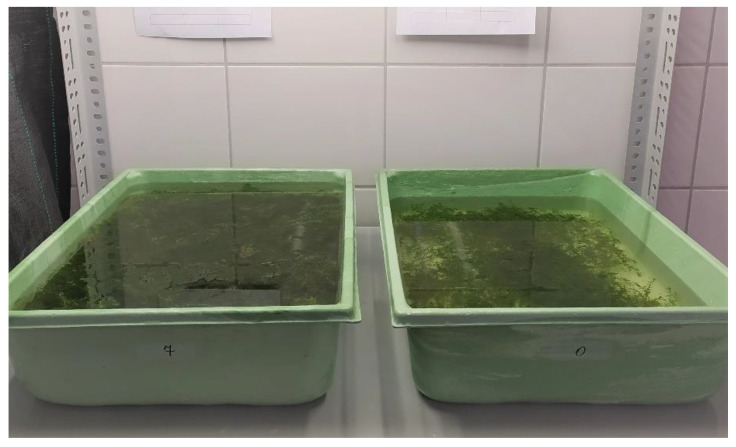
*Aldrovanda vesiculosa* plants during acclimatization to ex vitro conditions.

**Figure 4 biology-11-01389-f004:**
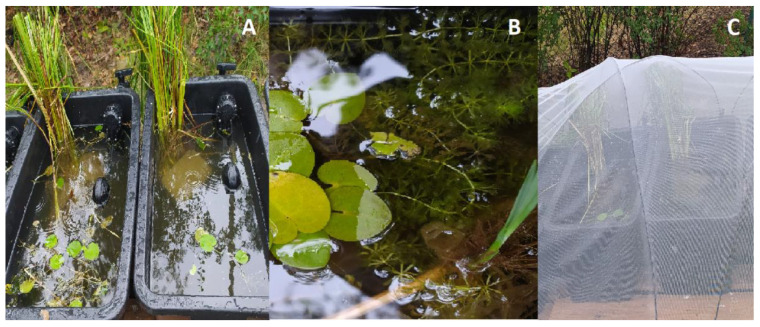
*Aldrovanda vesiculosa* plants in outdoor tanks; (**A**) tanks filled with lake water, visible clumps of grass and automatic water filling system; (**B**) floating aldrovanda shoots; (**C**) net covering tanks.

**Figure 5 biology-11-01389-f005:**
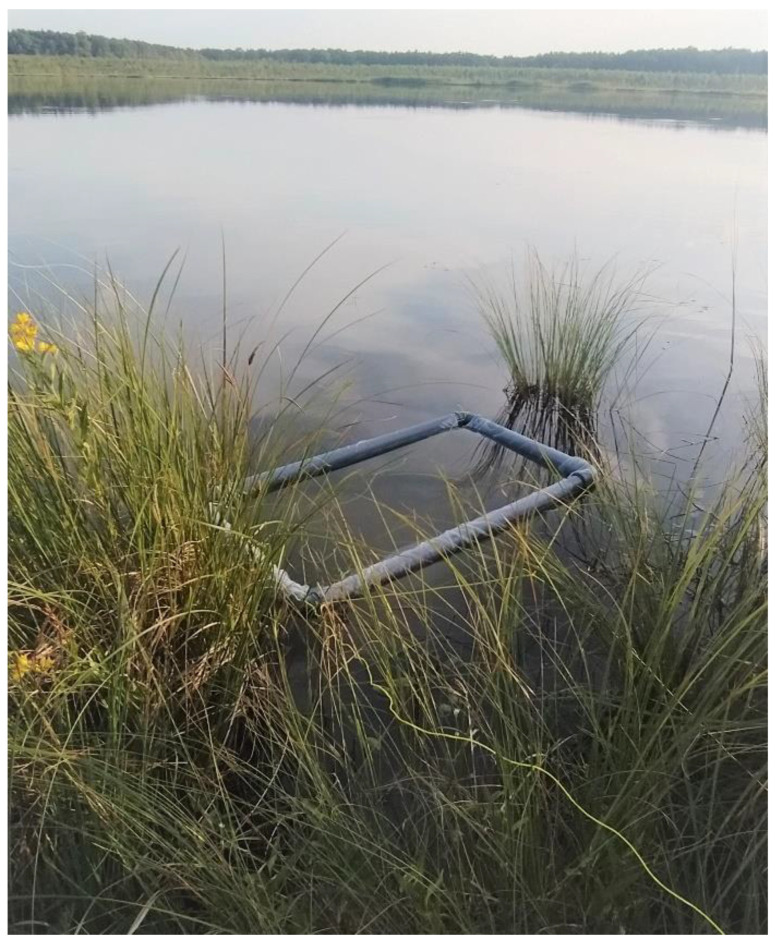
A net for monitoring of *Aldrovanda vesiculosa* plants in a natural lake.

**Figure 6 biology-11-01389-f006:**
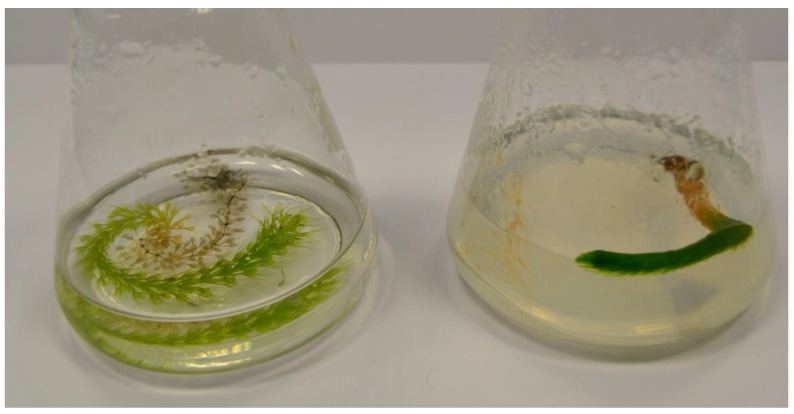
Regenerated shoots of *Aldrovanda vesiculosa* in liquid (left) and solidified (right) 1/5 MS media.

**Figure 7 biology-11-01389-f007:**
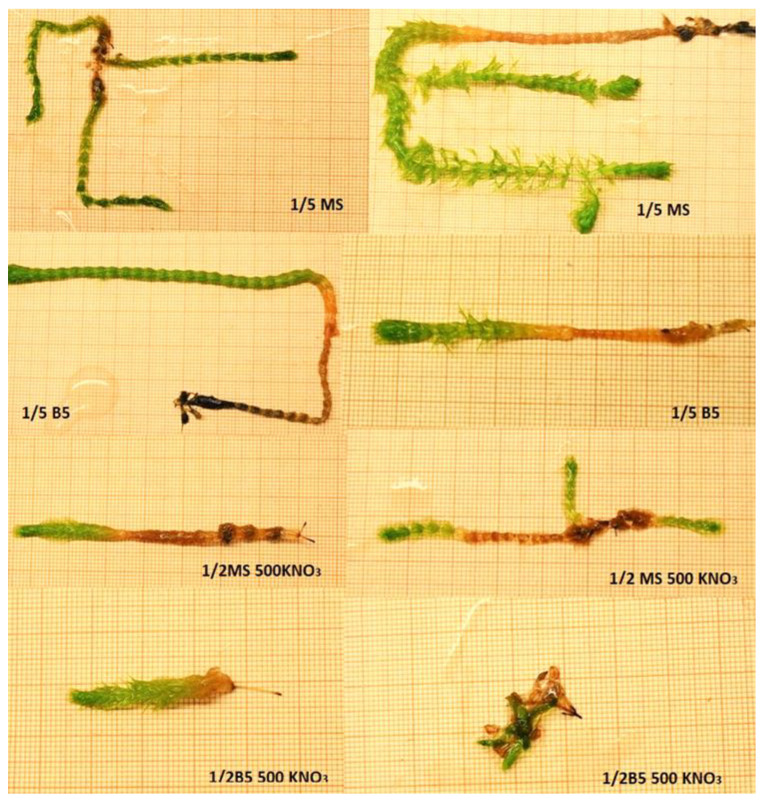
Shoots of *Aldrovanda vesiculosa* depending on the media type in tissue culture.

**Figure 8 biology-11-01389-f008:**
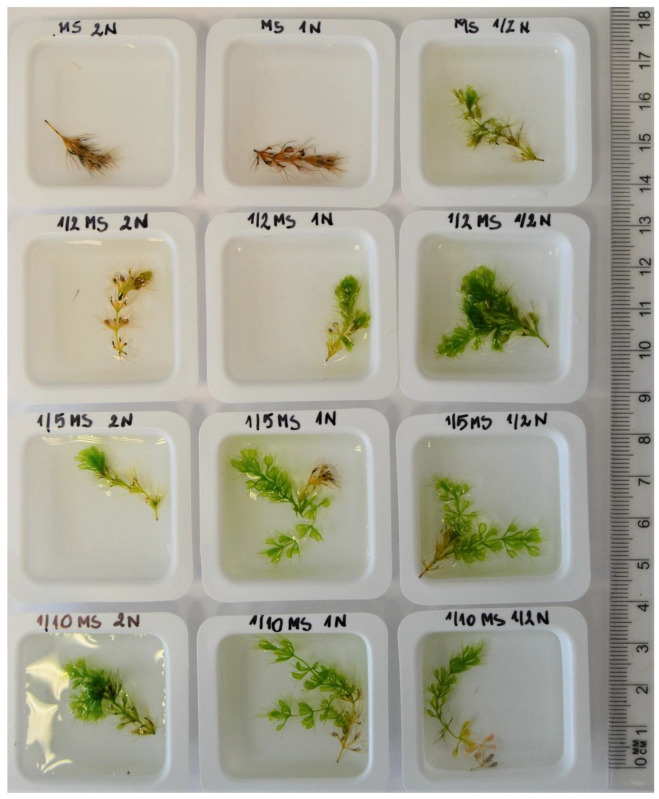
Shoots of *Aldrovanda vesiculosa* depending on the concentration of the media and nitrogen after 6 weeks in tissue culture. MS, 1/2MS, 1/5MS, 1/10MS represent the Murashige and Skoog medium concentration, 2N, 1N, and 1/2N represent the NH_4_NO_3_ + KNO_3_ concentration of the MS formulation.

**Figure 9 biology-11-01389-f009:**
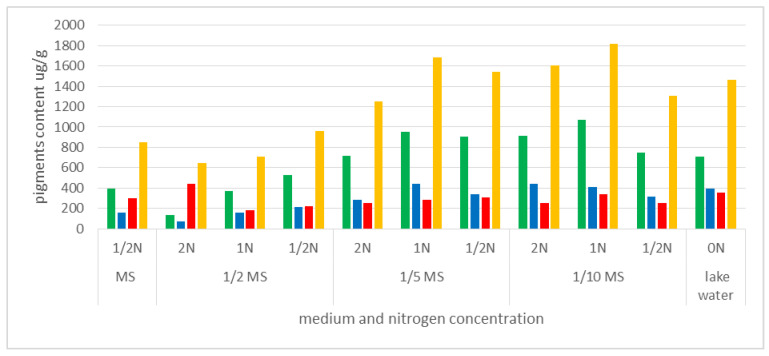
Content of photosynthetic pigments in *Aldrovanda vesiculosa* plant material from in vitro cultivation and from a natural site (Lake Łukie). MS—Murashige and Skoog medium formulation, 1/2N, 1/5MS, 1/10MS—mineral salts diluted to 1/2, 1/5, 1/10; N—content of nitrogen compounds NH_4_NO_3_ and KNO_3_ with concentration doubled (2N), unchanged (1N) or diluted twice (1/2N); ON—natural water lake. Green represents chlorophyll *a* content, blue—chlorophyll *b*, red—carotenoids, and yellow—sum of pigments.

**Figure 10 biology-11-01389-f010:**
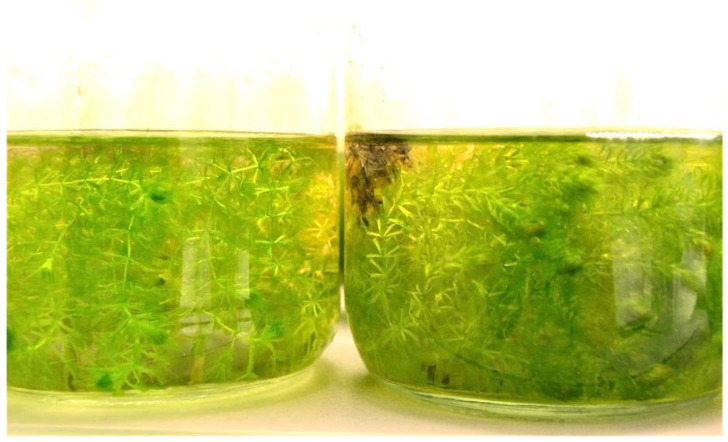
*Aldrovanda vesiculosa* shoots after 6 weeks in tissue culture (10 plants per 450 mL jar, with 200 mL liquid media).

**Table 1 biology-11-01389-t001:** Contamination and regeneration rate of *Aldrovanda vesiculosa* shoots initiated in vitro depending on medium solidification.

Media Type	Contaminated Explants (%)	Contamination-Free Explants (%)	Of Which:	
			Regenerating (%)	Necrotic (%)
Liquid	9	91	68	32
Solidified	13	87	3.5	96.5

**Table 2 biology-11-01389-t002:** Influence of medium type on the growth and development of *Aldrovanda vesiculosa* shoots in vitro in the first passage.

Medium	Regeneration (%)	Length of Main Shoot (mm)	Plants with Lateral Shoots (%)	Number of Lateral Shoots/Explant	Length of Lateral Shoots (mm)
1/5 MS	100%	76.80 a *	100 a	4.80 b	33.18 a
1/5 B5	60%	50.00 ab	33 a	10.0 a	17.40 b
1/2 MS 500 KNO_3_	60%	25.32 b	66.7 a	2.5 b	12.50 bc
1/2 B5 500 KNO_3_	60%	24.32 b	0 a	-	-

* Means followed by the same letter in columns do not differ significantly at α = 0.05. 1/5MS—Murashige and Skoog (MS) medium diluted 5 times, 1/5 B5—Gamborg B5 medium diluted 5 times, 1/2MS 500 KNO_3_—half-concentration MS medium with KNO_3_ concentration lowered to 500 mg·dm^−3^.

**Table 3 biology-11-01389-t003:** Influence of medium type on the growth and development of *Aldrovanda vesiculosa* shoots in vitro in the second passage.

	Length of Main Shoot (mm)	Length Increase (mm)	Number of Lateral Shoots/Shoot	Increase in Number of Lateral Shoots	Length of Lateral Shoots (mm)	Increase in Length of Lateral Shoots (mm)
1/5 MS	96.80 a *	20.0 a	6.2 ab	1.4 a	38.20 a	5.02 ab
1/5 B5	69.0 ab	19.0 ab	10.0 a	0 b	27.20 ab	9.80 a
1/2 MS 500 KNO_3_	27.32 b	2.0 b	2.5 b	0 b	14.60 bc	1.10 b
1/2 B5 500 KNO_3_	27.32 b	3.0 ab	2.0 b	2.0 a	11.0 c	0 b

* Means followed by the same letter in columns do not differ significantly at α = 0.05. 1/5MS—Murashige and Skoog (MS) medium diluted 5 times, 1/5 B5—Gamborg B5 medium diluted 5 times, 1/2MS 500 KNO_3_—half-concentration MS medium with KNO_3_ concentration lowered to 500 mg·dm^−3^.

**Table 4 biology-11-01389-t004:** Growth and development of *Aldrovanda vesiculosa* main shoots depending on the concentration of the media and nitrogen compounds.

MS Concentration	Nitrogen Concentration	Number of Regenerating Explants (%)	Main Shoot Length (mm)	Mean for Medium	Main Shoot Weight (mg)	Mean for Medium
MS	2N	0	-	-		-
	1N	0	-			
	1/2N	93	22.73 d *		32.63 ab	
1/2 MS	2N	45	20.5 d	23.56 C	20.18 b	31.86 A
	1N	95	22.24 d		29.39 b	
	1/2N	100	26.26 c		39.66 a	
1/5 MS	2N	97.5	22.56 d	27.96 B	18.18 c	26.15 BC
	1N	60	31.77 ab		31.84 ab	
	1/2N	100	29.50 bc		28.35 b	
1/10 MS	2N	100	30.85 ac	31.66 A	29.12 b	30.38 AB
	1N	100	35.64 a		32.27 ab	
	1/2N	100	28.49 bc		29.75 b	
	Mean for nitrogen concentration		2N—22.91 B 1N—27.61 A 1/2N—26.75 A		2N—20.64 C 1N—28.73 B 1/2N—32.62 A	

* Means followed by the same letter in columns do not differ significantly at α = 0.05. MS, 1/2MS, 1/5MS, 1/10MS represent the Murashige and Skoog medium concentration, 2N, 1N, and 1/2N represent the NH_4_NO_3_ + KNO_3_ concentration of the MS formulation.

**Table 5 biology-11-01389-t005:** Growth and development of *Aldrovanda vesiculosa* lateral shoots depending on the concentration of the media and nitrogen compounds.

MS Concentration	Nitrogen Concentration	Number of Explants with Lateral Shoots (%)	Number of Lateral Shoots	Mean for Medium	Length of Lateral Shoots (mm)	Mean for Medium	Weight of Lateral Shoots (mg)	Mean for Medium
MS	2N	-	-	1.15	-	8.18 B	-	13.75
	1N	-	-		-		-	
	1/2N	50.0	1.15 a *		8.18 ab		13.75 ab	
1/2 MS	2N	47.0	1.25 a	1.24	7.60 ab	8.03 B	5.62 ab	16.82
	1N	55.3	1.09 a		8.13 ab		12.76 ab	
	1/2N	44.7	1.41 a		8.12 ab		25.78 a	
1/5 MS	2N	25.6	1.10 a	1.48	6.64 ab	11.95	9.32 ab	19.29
	1N	60.0	1.50 a		12.61 a		23.36 a	
	1/2N	79.5	1.58 a		12.65 a		19.63 a	
1/10 MS	2N	53.8	1.05 a	1.11	13.27 a	13.70	13.69 ab	15.50
	1N	66.7	1.19 a		15.0 a		15.90 ab	
	1/2N	41.0	1.06 a		13.53 a		17.21 ab	
	Mean for nitrogen concentration		2N—0.87 1N—1.11 1/2N—1.35		2N—8.32 1N—10.78 1/2N—10.94		2N—8.82 1N—15.13 1/2N—19.14	

* Means followed by the same letter in columns do not differ significantly at α = 0.05. MS, 1/2MS, 1/5MS, 1/10MS represent the Murashige and Skoog medium concentration, 2N, 1N, and 1/2N represent the NH_4_NO_3_ + KNO_3_ concentration of the MS formulation.

## Data Availability

All the required data related to the current study are included in this manuscript.

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
