# Peer review of "Optimization of Propagation of the Polish Strain of Aldrovanda vesiculosa in Tissue Culture"

_biology, 2022, doi:10.3390/biology11101389_

Round 1
Reviewer 1 Report
This work is a very interesting application of micropoprpagatio to the conservation of an endangered species of relevant interest for polish biodiversity.
Author Response
Response to Review 1 comments
We, Authors of the article ‘The optimization of the propagation of Aldrovanda vesiculosa Polish strain in tissue culture’ would like to thank the Reviewer for the opinion on this manuscript.
We have applied the corrections suggested by the Reviewers. We are greatly thankful for the attention.
Yours sincerely,
On behalf of all co-authors
Magdalena Pogorzelec

Reviewer 2 Report
This could be an interesting story, however I could not understand much of the language used. The introduction is lengthy and much of it is not germane to the investigation I think?
Why 1/5 media used? Why 1/2 MS with nitrated. I did not see the rational to this. To be frank, I could not evaluate the topic of this manuscript with the language used.
Tables are very poorly constructed -- watch significant figures.
How many "samples per repetition" were used?
The language of the ms would need to be overhauled completely. It is full of English idioms (e.g. on the other hand) that are not best to use in scientific writings. The language needs to be evaluated by a native speaker before it can be considered for additional review.
To be honest, I could not follow much of the ms because of the poor language used. I realize that the authors are not native English speakers, but they should include an author or consultant that is proficient in English scientific writing. Only then can a reader understand exactly what the authors are trying to convey.
Author Response
Response to Review 2 comments
We, Authors of the article The optimization of the propagation of Aldrovanda vesiculosa Polish strain in tissue culture’ would like to thank the Reviewer for the opinion on this manuscript.
Our answers to the Reviewer are given below.
Review 2: This could be an interesting story, however I could not understand much of the language used.
Answer: The manuscript has been corrected by a native speaker.
Review 2: The introduction is lengthy and much of it is not germane to the investigation I think?
Answer: The introduction covers the most important issues, like the species itself, the problem of its extinction, and the use of tissue culture for plants species conservation.
Review 2: Why 1/5 media used? Why 1/2 MS with nitrated. I did not see the rational to this. To be frank, I could not evaluate the topic of this manuscript with the language used.
1/5 MS media was used as the basic media according to our previous preliminary research of aldrovanda and other Droseraceae species (not published).
Review 2: Tables are very poorly constructed -- watch significant figures.
We have constructed the tables in a simple way to present the values and the statistical data. These are typical tables for this type of research.
Review 2: How many "samples per repetition" were used?
The information has been added in the M&M part.
Review 2: The language of the ms would need to be overhauled completely. It is full of English idioms (e.g. on the other hand) that are not best to use in scientific writings. The language needs to be evaluated by a native speaker before it can be considered for additional review.
Review 2: To be honest, I could not follow much of the ms because of the poor language used. I realize that the authors are not native English speakers, but they should include an author or consultant that is proficient in English scientific writing. Only then can a reader understand exactly what the authors are trying to convey.
Answer: The manuscript has been corrected by a native speaker.
We hope the Reviewer will be satisfied with the significant changes we have made to the manuscript and that the information given in the manuscript is now presented in a better way.
Yours sincerely,
On behalf of all co-authors
Magdalena Pogorzelec

Reviewer 3 Report
The manuscript “The optimization of the propagation of Aldrovanda vesiculosa Polish strain in tissue culture” provides valuable and new information for the ex situ conservation of an endangered carnivorous species. Reports on the development of strategies to aid in the conservation of endangered plant species are extremely important within the scope of plant genetic resources conservation. However, despite the substantial amount of practical work performed to obtain the results presented in the manuscript, the optimization of the multiplication stage of a micropropagation protocol complemented with only a photosynthetic pigments study is not substantial enough for a full original article. The authors should consider, for instance, to develop the micropropagation protocol in its entirety (to obtain acclimatized plants for reintroduction in the wild), more studies on biological and biochemical parameters or even genetic studies to complement the work already developed and present a more significant manuscript. Besides, there is a recent publication by Adamec and Kondo (2022) (which is not cited in this manuscript), very similar to this manuscript. The authors must highlight their work from others that are like theirs using different approaches/studies to successfully publish their results.
Moreover, there are other issues that the authors should consider to improve the manuscript. The content and main ideas in the introduction section are adequate for the theme explored in the manuscript. However, the text is often confusing, some sentences are not well structured and there are English errors along the text (a few examples ahead). I suggest a proof reading from either a professional service or a native English speaker to reformulate the introduction section and improve the text on the other sections. Also, the literature chosen to support some statements are quite old and I suggest a partially update of the literature cited (examples ahead), including the recent in vitro studies for this species that are missing. Another important matter is to properly validate the endangered status of the species. Is A. vesiculosa identified as rare or endangered in a regional or national Red List or IUCN or other? If so, the authors should mention it in the text, a proof is recommended. As for the experimental procedure, the authors do not present a control medium when studying the influence of media type and nitrogen concentration.
Considering all the previous remarks, I cannot recommend this manuscript for publication in Biology.
Following please find more specific suggestions to improve the manuscript:
- Abstract – The authors should focus more on the media experiments results and photosynthetic pigments rather than the disinfection procedure.
- Line 20 – “…and the plants obtained in this way…” instead of “…and the obtained in this way plants…”
- Line 24-25 – “Aldrovanda in its natural habitat prefers low in nutrients, dystrophic waters.” – an example of a poorly structured sentence.
- Lines 34-36 – The authors should rearrange these initial sentences. Maybe start by saying that Aldrovanda is a genus containing only one species e then introduce the species. As it is, the authors first mention the species, then the genus (without stating which exact genus they´re referring to, is it Aldrovanda?) and back to the species name.
- Lines 34-39 – It should be clear where this species occur.
- Lines 48-49 – At which temperature can this species form seeds?
- Line 50 – “Adult plants may form up to 8 branches…” instead of “Adult plants may form even 8 branches…”.
- Line 58-59 – Very confusing sentence.
- Line 63 – What do the authors mean by “liter”? Is that the correct word? Don’t you mean something else?
- Line 65 – “…attached…”? Do the authors mean attracted?
- Line 65-66 – The references are old (2005 and 2012). To sustain this statement the authors should provide a more recent reference.
- Line 67-76 - Very confusing paragraph.
- Line 78 – “in situ” means in its natural site, it’s a repetition.
- Line 82-89 – The percentage of global plant species threatened with extinction seems very low, 8%. Again, the references are not very recent and since the authors refer to IUCN, they should investigate more accurate and recent percentages available on IUCN web site (or other recent source).
- Line 139 – The reference is from 2006, there are more recent references and the number of publications on ex situ conservation of endangered plant species has increased since then. Please rearrange this sentence in order to reflect the present state of this matter.
- Figures 1 and 2 – Consider merging these two figures as Figure 1a and 1b.
- Line 242 – “Aldrovanda vesiculosa is a plant species threatened with extinction” instead of “Aldrovanda vesiculosa is the plant species threatened with extinction”.
- Line 242-243 – “Even though Poland is the biggest reservoir of plants…” do the authors mean “the biggest reservoir of A. vesiculosa plants”? Please confirm throughout the text, there are many statements referring to plants in general, when the intention is to focus on carnivorous plants or this species.
- Line 247-249 – This statement is not entirely correct. Micropropagation allows to obtain large amounts of plant material in a short period of time (among other advantages) without harming wild/mother plants. However, to initiate the process, one needs to collect material from wild/mother plants, which implies harming the plants in this initial stage (unless seeds are used as starting material, which was not possible for A. vesiculosa).
- Table 1 – Should be presented only after been mentioned in the text.
- Line 271 – Please clarify in the text if these disinfected explants came both from disinfected shoots and turions.
- Table 2 – Is not mentioned in the text. Why there is no statistical analysis for the parameter “Plants with lateral shoots”?
- Table 3 – What do the columns “Change” mean in the table or to what they’re referring to? And provide an explanation in the text.
- Table 4 and 5 – Why did the authors performed this type of statistical analysis by “Mean a” e “Mean b”? The text is also not in alignment with this analysis.
- Figure 7 - Is not mentioned in the text.
- Line 363-364 – Verify the reference, a parenthesis seems to be missing.
- Line 375-379 – The authors should explore better the relation between their results, which were better in low nutrient media, and the type of plant studied, carnivorous plant, and its nutrient necessities in the wild.
- Line 398-400 – This might be valid for carnivorous plants only. There are numerous studies concerning photosynthetic parameters in plants. Reformulate the sentence to be accurate.
- Figure 8 – Does green bars correspond to chlorophyll a? And there is no statistical analysis.
- Conclusion – Like the abstract, much emphasis on disinfection and little on culture media and pigments.
- All manuscript - Standardize units, be consistent throughout the text (for example line 413 is mg dm-3 and on line 416 is µg/g).
Author Response
Response to Review 3 comments
We, Authors of the article The optimization of the propagation of Aldrovanda vesiculosa Polish strain in tissue culture’ would like to thank the Reviewer for the opinion on this manuscript.
Our answers to the Reviewers’ comments and suggestions are given below.
Review 3: The manuscript “The optimization of the propagation of Aldrovanda vesiculosa Polish strain in tissue culture” provides valuable and new information for the ex situ conservation of an endangered carnivorous species. Reports on the development of strategies to aid in the conservation of endangered plant species are extremely important within the scope of plant genetic resources conservation. However, despite the substantial amount of practical work performed to obtain the results presented in the manuscript, the optimization of the multiplication stage of a micropropagation protocol complemented with only a photosynthetic pigments study is not substantial enough for a full original article. The authors should consider, for instance, to develop the micropropagation protocol in its entirety (to obtain acclimatized plants for reintroduction in the wild), more studies on biological and biochemical parameters or even genetic studies to complement the work already developed and present a more significant manuscript.
Answer: We have added the information considering the acclimatization and reintroduction of the obtained plants.
Review 3: Besides, there is a recent publication by Adamec and Kondo (2022) (which is not cited in this manuscript), very similar to this manuscript. The authors must highlight their work from others that are like theirs using different approaches/studies to successfully publish their results.
Answer: We have cited all known to us literature. I have contantec Mr Adamec personally, and there is no publication released in the recent years on aldrovanda propagation. This is probably the article cited in the manuscript, dated 1997 and given in the Research Gate in 2022.
Review 3: Moreover, there are other issues that the authors should consider to improve the manuscript. The content and main ideas in the introduction section are adequate for the theme explored in the manuscript. However, the text is often confusing, some sentences are not well structured and there are English errors along the text (a few examples ahead). I suggest a proof reading from either a professional service or a native English speaker to reformulate the introduction section and improve the text on the other sections.
Answer: The manuscript has been corrected by a native speaker.
Review 3: Also, the literature chosen to support some statements are quite old and I suggest a partially update of the literature cited (examples ahead), including the recent in vitro studies for this species that are missing. Another important matter is to properly validate the endangered status of the species. Is A. vesiculosa identified as rare or endangered in a regional or national Red List or IUCN or other? If so, the authors should mention it in the text, a proof is recommended.
Answer: We have added the information.
Review 3: As for the experimental procedure, the authors do not present a control medium when studying the influence of media type and nitrogen concentration.
Answer: The 1/5 MS or 1/10 MS media can be considered as the control ones. We think there is no sense to use a typical control (MS or Gamborg media) because the concentration of the elements is too strong for aldrovanda.
Review 3: Abstract – The authors should focus more on the media experiments results and photosynthetic pigments rather than the disinfection procedure.
Answer: There is very little information regarding micropropagation of the aldrovanda and it is given in the discussion part.
Review 3: Line 20 – “…and the plants obtained in this way…” instead of “…and the obtained in this way plants…”
- Line 24-25 – “Aldrovanda in its natural habitat prefers low in nutrients, dystrophic waters.” – an example of a poorly structured sentence.
- Lines 34-36 – The authors should rearrange these initial sentences. Maybe start by saying that Aldrovanda is a genus containing only one species e then introduce the species. As it is, the authors first mention the species, then the genus (without stating which exact genus they´re referring to, is it Aldrovanda?) and back to the species name.
Answer: The manuscript has been corrected by a native speaker.
Review 3: Lines 34-39 – It should be clear where this species occur.
Answer: The information is given.
Review 3: Lines 48-49 – At which temperature can this species form seeds?
Answer: The information has been added.
Review 3: Line 50 – “Adult plants may form up to 8 branches…” instead of “Adult plants may form even 8 branches…”.
- Line 58-59 – Very confusing sentence.
- Line 63 – What do the authors mean by “liter”? Is that the correct word? Don’t you mean something else?
- Line 65 – “…attached…”? Do the authors mean attracted?
Answer: As above, the manuscript has been corrected.
Review 3: Line 65-66 – The references are old (2005 and 2012). To sustain this statement the authors should provide a more recent reference.
- Line 67-76 - Very confusing paragraph.
- Line 78 – “in situ” means in its natural site, it’s a repetition.
- Line 82-89 – The percentage of global plant species threatened with extinction seems very low, 8%. Again, the references are not very recent and since the authors refer to IUCN, they should investigate more accurate and recent percentages available on IUCN web site (or other recent source).
- Line 139 – The reference is from 2006, there are more recent references and the number of publications on ex situ conservation of endangered plant species has increased since then. Please rearrange this sentence in order to reflect the present state of this matter.
Answer: The manuscript has been corrected.
Review 3: Figures 1 and 2 – Consider merging these two figures as Figure 1a and 1b.
Answer: The Figures have been merged.
Review 3: Line 242 – “Aldrovanda vesiculosa is a plant species threatened with extinction” instead of “Aldrovanda vesiculosa is the plant species threatened with extinction”.
Answer: corrected
Review 3: Line 242-243 – “Even though Poland is the biggest reservoir of plants…” do the authors mean “the biggest reservoir of A. vesiculosa plants”? Please confirm throughout the text, there are many statements referring to plants in general, when the intention is to focus on carnivorous plants or this species.
Answer: It has been corrected.
Review 3: Line 247-249 – This statement is not entirely correct. Micropropagation allows to obtain large amounts of plant material in a short period of time (among other advantages) without harming wild/mother plants. However, to initiate the process, one needs to collect material from wild/mother plants, which implies harming the plants in this initial stage (unless seeds are used as starting material, which was not possible for A. vesiculosa).
Answer: The Reviewer is right. It has been corrected.
Review 3: Table 1 – Should be presented only after been mentioned in the text.
Answer: It has been corrected.
Review 3: Line 271 – Please clarify in the text if these disinfected explants came both from disinfected shoots and turions.
- Table 2 – Is not mentioned in the text. Why there is no statistical analysis for the parameter “Plants with lateral shoots”?
Answer: There are no significant differences, we have added ‘a’ with all the values.
Review 3: Table 3 – What do the columns “Change” mean in the table or to what they’re referring to? And provide an explanation in the text.
Answer: It has been changed.
Review 3: Table 4 and 5 – Why did the authors performed this type of statistical analysis by “Mean a” e “Mean b”? The text is also not in alignment with this analysis.
Answer: This type of analysis shows if the media itself or concentration of nitrogen had a significant influence on the measured parameters.
Review 3: Figure 7 - Is not mentioned in the text.
Answer: The figures have been added to the manuscript and all of them are mentioned in the text.
Review 3: Line 363-364 – Verify the reference, a parenthesis seems to be missing.
Answer: The references has been checked and corrected.
Review 3: Line 375-379 – The authors should explore better the relation between their results, which were better in low nutrient media, and the type of plant studied, carnivorous plant, and its nutrient necessities in the wild.
Answer: The information has been added.
Review 3: Line 398-400 – This might be valid for carnivorous plants only. There are numerous studies concerning photosynthetic parameters in plants. Reformulate the sentence to be accurate.
Answer: corrected.
Review 3: Figure 8 – Does green bars correspond to chlorophyll a? And there is no statistical analysis.
Answer: The caption has been corrected.
Review 3: Conclusion – Like the abstract, much emphasis on disinfection and little on culture media and pigments.
Answer: It has been changed.
Reivew 3: All manuscript - Standardize units, be consistent throughout the text (for example line 413 is mg dm-3 and on line 416 is µg/g).
Answer: The units has been changed.
We hope the Reviewer will be satisfied with the significant changes we have made to the manuscript.
Yours sincerely,
On behalf of all co-authors
Magdalena Pogorzelec

Reviewer 4 Report
In this exciting article, the generation of tissue cultures of the endangered waterwheel plant "Aldrovanda vesiculosa" is described. A detailed introduction to the plant is followed by the material and methods used, results, and outlook, which provide a good guide to the in vitro cultivation of the Polish strain of A. vesiculosa. The submitted article was incredibly exciting to me because, like many of my colleagues, I have been primarily involved with the in vitro cultivation of terrestrial plants. I am therefore convinced that it will also be of great interest to other readers.
Unfortunately, however, it cannot yet be published in its present form, as some aspects still need to be worked out more clearly. Furthermore, some linguistic and formal deficiencies still must be eliminated. I, therefore, recommend a revision of the article. In the following, some points are listed:
Comments:
1. If you all work in the same university, you can use one affiliation. If there are several groups involved, please indicate them.
2. Abstract (lines 16 to 29): The abstract contains an introduction and method part, but in my opinion, the results are a little bit neglected. Please elaborate on them more so that an interested colleague can immediately see what the benefits of your publication are. And you may delete sentences like “The optimal media for the plants micropropagation should contain all the necessary elements.” as this is not a new finding.
3. Keywords (lines 28 to 29): You may add some more keywords which are special for your approach, e.g. turion sterilization
4. Figures 1 and 2: I really like the figures, but you should combine them into one figure (as you have done with figure 4).
5. Line 171: “containing the 1/5 MS media, which consisted of the 10 times diluted Murashige and Skoog (MS)”. I do not get the point here. Is it 1/10 MS or 1/5 MS?
6. Line 175: The pH was set to 5.5 prior to autoclaving. Have you measured the pH afterward, as it usually decreases due to the sucrose cleaving? What was the pH of the lake water you took the plants from? What is the beneficial pH range for growing Aldrovanda?
7. Line 177: Depending on the form of thiamine (chloride, hydrochloride, etc.) and the pH, negative effects of autoclaving have been reported. Which form of thiamine (line 172) have you used?
8. Line 179: You mentioned (lines 153 to 154) that you sampled the turions in January and February, and here you mentioned they were cultivated in March. Can you specify the time they were kept in storage before you recultivated them (e.g. in days)? This would make it easier to duplicate your findings.
9. Line 229: Please indicate the rcf rather than the rpm, as this makes it easier to reproduce your findings using other centrifuges.
10. Table 1 (line 264): I had expected the sum of regenerating explants, necrotic explants, and contaminated explants would sum up to 100%. Also, please indicate the number of explants you used.
11. Table 2 (line 294): Please indicate the number of explants you used for shoot generation.
12. Line 384: The Murashige and Skoog Medium (MS) abbreviation is introduced at the beginning of the article, but the name is often written out again. In my opinion, this is not necessary for the most commonly used tissue culture medium. However, at this point, you mention the rather uncommon RM medium, but the abbreviation is not explained anywhere. Please write out the RM medium name here.
Formal and spelling mistakes (the list is not exhaustive):
1. Line 189: temperaturÄ™
2. Line 214: use of protected spaces, “450 ml”, the same with line 229 “14 000 rpm”
3. Line 302: 76,80 mm should be 76.80 mm
4. Table 5 (line 351): Aldrovanda vesiculosa should be in italics
5. Table 5, bottom: different dashes for 2N – 0.87 and 1N - 1.11
6. Line 385: Wrongly capitalization of “Depending”
7. Line 416: Here you use µg/g, whereas you mostly use the mg·dm-3 style. Please stick to one style
Please read your manuscript again carefully, as it contains several errors, and the grammar in some places makes it difficult to understand the text.
I look forward to reading the revised version of this interesting article and wish you success in correcting it.
Author Response
Response to Review 4 comments
We, Authors of the article The optimization of the propagation of Aldrovanda vesiculosa Polish strain in tissue culture’ would like to thank the Reviewer for the opinion on this manuscript.
Our answers (in black) to the Reviewers’ comments and suggestions (in red) are given below.
Review 4: Unfortunately, however, it cannot yet be published in its present form, as some aspects still need to be worked out more clearly. Furthermore, some linguistic and formal deficiencies still must be eliminated. I, therefore, recommend a revision of the article.
The manuscript has been corrected by a native speaker.
Review 4: If you all work in the same university, you can use one affiliation. If there are several groups involved, please indicate them.
It has been corrected.
Review 4: Abstract (lines 16 to 29): The abstract contains an introduction and method part, but in my opinion, the results are a little bit neglected. Please elaborate on them more so that an interested colleague can immediately see what the benefits of your publication are. And you may delete sentences like “The optimal media for the plants micropropagation should contain all the necessary elements.” as this is not a new finding.
The abstract has been changed.
Review 4: Keywords (lines 28 to 29): You may add some more keywords which are special for your approach, e.g. turion sterilization
A few additional keywords have been added.
Review 4: Figures 1 and 2: I really like the figures, but you should combine them into one figure (as you have done with figure 4).
The figures have been combined into one.
Review 4: Line 171: “containing the 1/5 MS media, which consisted of the 10 times diluted Murashige and Skoog (MS)”. I do not get the point here. Is it 1/10 MS or 1/5 MS?
It has been corrected.
Review 4: Line 175: The pH was set to 5.5 prior to autoclaving. Have you measured the pH afterward, as it usually decreases due to the sucrose cleaving? What was the pH of the lake water you took the plants from? What is the beneficial pH range for growing Aldrovanda?
The pH was similar to the pH we use for Drosera sp. and according to our observations. We have not checked the pH after autoclaving. The pH of the lake water we took the plants from was around 6.
Review 4: Line 177: Depending on the form of thiamine (chloride, hydrochloride, etc.) and the pH, negative effects of autoclaving have been reported. Which form of thiamine (line 172) have you used?
We have used the hydrochloride form of thiamine.
Review 4: Line 179: You mentioned (lines 153 to 154) that you sampled the turions in January and February, and here you mentioned they were cultivated in March. Can you specify the time they were kept in storage before you recultivated them (e.g. in days)? This would make it easier to duplicate your findings.
The information has been added.
Review 4: Line 229: Please indicate the rcf rather than the rpm, as this makes it easier to reproduce your findings using other centrifuges.
Both values are given.
Review 4: Table 1 (line 264): I had expected the sum of regenerating explants, necrotic explants, and contaminated explants would sum up to 100%. Also, please indicate the number of explants you used.
There is 100%. The table has been changed to make it more clear.
Review 4: Table 2 (line 294): Please indicate the number of explants you used for shoot generation.
The information is given in the M&M.
Review 4: Line 384: The Murashige and Skoog Medium (MS) abbreviation is introduced at the beginning of the article, but the name is often written out again. In my opinion, this is not necessary for the most commonly used tissue culture medium. However, at this point, you mention the rather uncommon RM medium, but the abbreviation is not explained anywhere. Please write out the RM medium name here.
According to the Journal requirements, it should be written in full name when mentioned for the first time in the introduction, methods and results sections. The RM medium is mentioned in the results and discussion part. It is very uncommon and very difficult to find, while the MS medium is known for everyone.
Review 4:
Formal and spelling mistakes (the list is not exhaustive):
- Line 189: temperaturÄ™
- Line 214: use of protected spaces, “450 ml”, the same with line 229 “14 000 rpm”
- Line 302: 76,80 mm should be 76.80 mm
- Table 5 (line 351): Aldrovanda vesiculosa should be in italics
- Table 5, bottom: different dashes for 2N – 0.87 and 1N - 1.11
- Line 385: Wrongly capitalization of “Depending”
- Line 416: Here you use µg/g, whereas you mostly use the mg·dm-3style. Please stick to one style
The above and more mistakes have been corrected. The template does not allow to use protected spaces.
Please read your manuscript again carefully, as it contains several errors, and the grammar in some places makes it difficult to understand the text.
The manuscript has been corrected by a native speaker.
We hope the Reviewer will be satisfied with the significant changes we have made to the manuscript.
Yours sincerely,
On behalf of all co-authors
Magdalena Pogorzelec

Round 2
Reviewer 2 Report
Thank you for the changes. To be honest, the language is improved, but many errors still remain. Specifically, the use of English articles in scientific writing is very different than general English composition. Many of the sentences are very wordy; conciseness in scientific writing is important. The arrangement of some of the sentences is still poor (subject is buried deep within the sentence -- avoid long introductory phrases) and there are some tense changes. I suggest a professional scientific editor as the "native speaker" may not provide the best result for a "science paper".
Significant figures in Tables are still not correct. I doubt that you measured structures to hundredth of mm. I suggest that you use whole numbers for this. e.g, Table 5 Number of explants with lateral shoots, lateral shoots, length all should be whole numbers. You are presenting arithmetic means that are not appropriate.
The unpublished data concerning media composition should be included in the ms. It is somewhat unsatisfying to say in another unpublished study we found. . . ..
Reviewer 3 Report
The reviewers’ concerns and suggestions were in general properly addressed by the authors.
The inclusion of a “Acclimatization and reintroduction process” improved the manuscript and highlighted the necessity for preserving this species. The English was substantially improved throughout the manuscript. The information about the endangered status of the species A. vesiculosa was also included. Still, there are a few improvements to be done, including the use of more updated references in some parts of the manuscript. Overall, with a minor revision, I can recommend this paper for publication on Biology.
Following please find a few more specific suggestions to improve the manuscript:
Line 17 – “Contamination-free in vitro cultures were established…” instead of “Contamination-free in vitro cultivation was established…”.
All manuscript – Remove the “the” before medium, when referring to a specific medium. For example, line 19 “The explants regenerated better in the liquid 1/5 MS medium than in the solidified medium.”.
Line 23 – Suggestion: Remove “…in the fresh weight of leaves…” or rearrange the sentence.
Line 58 – “Falling leaf litter…”.
Line 75 – A coma is missing: “[7,24…”.
Line 92-94 – In vitro propagation is a short-term storage/preservation ex situ technique or medium-term if slow growth is applied. Long-term storage/preservation ex situ techniques refer to cryopreservation. The authors should either remove that part or rearrange the sentence resorting to a more recent reference.
Line 102 – There are more recent citations referring to biotechnological techniques used for plant conservation, including endangered species.
Line 103 – Consider changing this sentence for example to: “Tissue culture is usually initiated in the laboratory using plant material collected in the field”.
Line 108-109 – Contamination is the main problem in the initiation phase of in vitro culture, not in vitro culture in general.
Line 120 – Use the term “acclimatization” instead of “weaning”.
Line 185 – The reference for Gamborg medium is correct? Is it not reference 59? Anyhow, reference 59 is not in the text. The authors should make a thorough reference check.
Line 207 – “The fresh plants were also analysed…”
Line 259 – “As the method of sterilization…”
Line 270 – Use the abbreviation MS, it was already mentioned before in the Material and Methods section.
Line 289-290 – Suggestion: “…the media type on the in vitro growth of A. vesiculosa shoots in vitro.” or “…the media type on A. vesiculosa shoots growing in vitro.”
Figure 9 – Still there is no statistical analysis nor explanation was given for that.
Subsection 3.5 – Do the authors have any data concerning the percentage of plants that were successfully acclimatized to ex vitro conditions? Was it 100%? That information should be included here. Also, if there is any available data about size or other parameter, it would be interesting to include here (and refer in the Material and Methods how those measures/calculations were performed).
Line 439 – Suggestion: “We present the a micropropagation protocol method for the Polish strain of A. vesiculosa.”
Line 446-447 – “…large-scale micropropagation of A. vesiculosa shoots in vitro.” or “…large-scale in vitro propagation of A. vesiculosa shoots in vitro.”.
Reviewer 4 Report
Thank you for the good and comprehensive revision. The manuscript reads much better and the newly inserted parts are very interesting.
I would still recommend the authors to have the text proofread again, to split some of the longer sentences into several, and to expand the labels of the tables and figures (for example, the description of the newly inserted Figure 4 is very short and does not go into details considering parts A, B, and C).
